# Parent-Reported Changes in Ontario Children’s Physical Activity Levels during the COVID-19 Pandemic

**DOI:** 10.3390/children10020221

**Published:** 2023-01-26

**Authors:** Monika Szpunar, Matthew Bourke, Leigh M. Vanderloo, Brianne A. Bruijns, Stephanie Truelove, Shauna M. Burke, Jason Gilliland, Jennifer D. Irwin, Patricia Tucker

**Affiliations:** 1Health and Rehabilitation Sciences, Faculty of Health Sciences, University of Western Ontario, London, ON N6G 1H1, Canada; 2School of Occupational Therapy, Faculty of Health Sciences, University of Western Ontario, London, ON N6G 1H1, Canada; 3ParticipACTION, 77 Bloor Street West, Suite 1205, Toronto, ON M5S 1M2, Canada; 4Member Interest Groups Section, Professional Development and Practice Support, College of Family Physicians of Canada, Mississauga, ON L4W 5A4, Canada; 5School of Health Studies, Faculty of Health Sciences, University of Western Ontario, London, ON N6G 1H1, Canada; 6Children’s Health Research Institute, Lawson Health Research Institute, London, ON N6C 2R5, Canada; 7Department of Geography and Environment, University of Western Ontario, London, ON N6G 1H1, Canada; 8Department of Pediatrics, University of Western Ontario, London, ON N6G 1H1, Canada; 9Department of Epidemiology & Biostatistics, University of Western Ontario, London, ON N6G 1H1, Canada

**Keywords:** physical activity, COVID-19, children, Ontario, pandemic

## Abstract

The COVID-19 pandemic resulted in closures of physical-activity-supporting environments, including playgrounds, outdoor recreation facilities (e.g., basketball courts), and community centers, which impacted children’s movement opportunities. This study evaluated changes in Ontario children’s physical activity levels during the COVID-19 pandemic and explored the impact of family sociodemographic markers on children’s activity. Parents (*n* = 243; *M_age_* = 38.8 years) of children aged 12 and under (*n* = 408; *M_age_* = 6.7 years) living in Ontario, Canada, completed two online surveys between August and December 2020 (survey 1) and August and December 2021 (survey 2). Generalized linear mixed-effects models were used to estimate changes in the proportion of children who accumulated 60 min of physical activity per day pre-lockdown, during lockdown, and post-lockdown in Ontario. Results revealed a significant non-linear trajectory whereby the proportion of children achieving 60 min of physical activity per day pre-lockdown (63%) declined during lockdown (21%) and then increased post-lockdown (54%). Changes in the proportion of children engaging in 60 min of daily physical activity were moderated by several demographic variables. Efforts are needed to provide parents of young children with a wider variety of resources to ensure children are obtaining sufficient levels of physical activity regardless of the presence of community lockdowns.

## 1. Introduction

The health benefits associated with meeting international (i.e., World Health Organization [WHO]) [1] and national (e.g., United Kingdom [2], Australia [3], and Canada [4]) physical activity recommendations are well documented [5]. For children, participating in recommended levels of daily physical activity is associated with improved cardiometabolic health [6], lower risk of obesity [5], improved mental health and wellbeing [7], and reduced risk of chronic disease in adolescence and later in life [8]. Consequently, both the Canadian 24-Hour Movement Guidelines [9] and the WHO Physical Activity Guidelines [1] suggest that infants (<1 year) should engage in at least 30 min of a variety of physical activities (including tummy time) [9], while toddlers (1–2 years) and preschoolers (3–4 years) are encouraged to engage in 180 min of a variety of physical activities (whereby 60 min should be at the moderate-to-vigorous intensity for 3-to-4-year-olds [9]). Further, it is recommended that school-aged children (5–17 years) strive for a minimum of 60 min of moderate-to-vigorous physical activity (MVPA) per day [10,11]. However, recent data from the Canadian Health Measures Survey (CHMS) suggest that only 62% of toddlers and preschoolers and 47% of school-aged children aged (5–11 years) are meeting physical activity recommendations [12,13]. These low levels of physical activity are not unique to Canada; globally, approximately only one-third of children between the ages of 5–17 years are achieving their respective national physical activity guidelines [14]. 

Coronavirus disease 2019 (COVID-19) was announced a global pandemic by the WHO on 11 March 2020 [15]. Shortly thereafter, stay-at-home orders (e.g., lockdowns) were implemented and resulted in restrictions on access to public spaces. In addition, community-wide public health protections were implemented by the Ontario government in accordance with the national COVID-19 strategy in Canada [16]. For parents and guardians, these protections drastically reduced opportunities for their children to engage in physical activities [17,18], as many physical activity-supporting environments (e.g., schools, childcare, and community centers, including indoor and outdoor sport recreation facilities) were inaccessible for extended periods of time [16]. Further, access to outdoor public play spaces such as parks and basketball courts was also prohibited during the early stages of COVID-19 in Ontario due to transmission risks [19]. These public health protections, in the form of lockdowns, remained in place for extended periods (in a staggered format, dependent on positive case counts) in Ontario, Canada [20], leaving many families with limited options for engaging in physical activity-related behaviors outside of their homes.

Despite existing targets, the prevalence of Canadian children meeting the movement guidelines pre-pandemic was low [12,21,22] and has further declined since March 2020 [23]. A study conducted by Moore and colleagues during the early stages of the pandemic (i.e., April 2020) identified that less than one-fifth (18.2%) of Canadian children (aged 5–11 years; *n* = 1472), as reported by their parents, were meeting the movement behavior guidelines [23]. Six months later (i.e., October 2020), this number declined to 14.3% among the same group of children [24]. Given the observed decline of physical activity among Canadian children during the COVID-19 pandemic—and as the pandemic continues—it is imperative to explore how children’s activity levels have been impacted and what factors may have influenced their physical activity participation during the COVID-19 pandemic. For example, research conducted during the pandemic has found various family sociodemographic factors (e.g., dog ownership, presence of outdoor space at home) as important correlates of children’s physical activity [23,25]. Given the lack of published evidence pertaining to the impact of the pandemic on young children’s (0–5 years) movement behaviors, the purpose of this study was to explore changes in Ontario children’s (0–12 years) physical activity levels during the first 1.5 years of the COVID-19 pandemic. Specifically, children’s physical activity levels prior to the declaration of a pandemic and lockdown (*before March 2020; measured retrospectively*), during lockdowns (*March 2020–June 2020; January 2021–May 2021),* and post-lockdown *(between August to December 2021)* were captured (At *post-lockdown*, stay-at-home periods in Ontario ceased and various sport settings were permitted to re-open. For more information, a timely breakdown of COVID-19 restriction easing (e.g., settings that were open for operation) can be found here: https://www.jdsupra.com/legalnews/ontario-s-covid-19-response-a-history-1280608/ accessed on 29 April 2022). For the purpose of this study, lockdown refers to periods of time in Ontario when citizens were required to stay at home, and public spaces (e.g., gyms, schools, and community centers), were closed. A secondary objective was to examine the impact of family sociodemographic and environmental moderators such as housing type, children’s biological sex and age, annual family household income, and parental level of education on physical activity.

## 2. Materials and Methods

### 2.1. Study Design and Procedures

Drawing on data collected from the larger *Return to Play* study (details published elsewhere [26]), this repeated measures study consisted of two online surveys (August–December 2020 (survey 1); and August–December 2021 (survey 2)) designed to assess parents’ plans for their children’s return to play/sport over the course of the COVID-19 pandemic. Data concerning parents’ and children’s physical activity levels were also collected. The current paper presents parent-reported data collected in both surveys related to their children’s physical activity levels throughout the pandemic (including retrospectively, to capture pre-pandemic levels and during periods of lockdown). See Figure 1 for a timeline of the *Return to Play* study surveys and provincial announcements (e.g., COVID-19 provincial mandates) and Appendix A for items from the *Return to Play* Surveys used for this paper. Ethical approval was provided by the Non-Medical Research Ethics Board at the University of Western Ontario (REB #116331), and informed consent was obtained from all participants.

### 2.2. Recruitment and Participants

To recruit participants, the research team contacted various child sport and physical activity organizations across Ontario and invited them to share study details (e.g., infographic with survey link) with their communities. In addition, multiple social media platforms (e.g., Instagram and Facebook) were used. English-proficient Ontario parents and guardians with children aged 12 and under (at the time of baseline survey completion) were invited to participate. Upon confirmation of study eligibility, participants were directed to the first online survey (the completion of which indicated their consent to participate). At the end of the first online survey, participants were asked to provide their email address to be contacted for subsequent surveys. 

### 2.3. Instruments and Tools

Two online surveys were created by the research team, and the Tolerance for Risk in Play Scale (TRiPS) validated tool was incorporated into the baseline survey [27]. The surveys were administered in English and delivered via Qualtrics and took approximately 30 min to complete. Survey items were informed by the COVID-19 situation in Ontario, Canada, at the time of survey creation (i.e., survey 1, launched August 2020, *n* = 162 items; and survey 2, launched August 2021, *n* = 58 items). Survey items were embedded into sections based on type of question asked (e.g., demographics and physical activity levels), and unique instructions were provided based on each section. For this paper, sociodemographic items known to be associated with children’s physical activity (e.g., dog ownership, parents’ level of education; *n* = 16 [23,28]) and parent-reported physical activity levels of their children pre-lockdown assessed in the baseline survey, and items that assessed parent-reported physical activity levels of children during lockdown and post-lockdown from the follow-up survey were examined.

### 2.4. Sociodemographic Questions

Sociodemographic information (*n* = 16 items) such as parent’s age, self-report gender, and highest level of education, as well as parent-reported child biological sex, age, and disability status, were measured. Participants were asked to report the number of children aged 12 and under they cared for at the time of survey completion (i.e., *How many children aged 0–12 years do you currently provide care for?*) via a dropdown list (i.e., participants could select any number). As such, the questions that followed were populated based on the participants’ selected number of children. For example, if a participant selected that they have 2 children under 12 years of age, they had to fill out age and biological sex twice, once per each of their children. Additional items captured socio-economic factors such as employment status, family household income, housing type (e.g., apartment, detached home), and community type (e.g., rural, urban), as well as dog ownership.

### 2.5. Children’s Physical Activity

In Survey 1, parents reported their children’s daily physical activity levels pre-lockdown. Specifically, the survey asked, *“In your opinion, how many minutes per day did your child spend engaged in physical activity prior to the COVID-19 pandemic?*” Response options were presented in multiple-choice format, with four response options (i.e., less than 30 min per day, 30–59 min per day, 60–149 min per day, or 150 min or more per day), and participants were required to input a value for each of their children aged 12 and under.

At follow-up (i.e., Survey 2), parents responded to two questions concerning each of their children’s physical activity during lockdowns and post-lockdown. The questions asked, “*In your opinion, how many minutes per day did your child spend engaged in physical activity during Ontario’s strictest COVID-19 related lockdowns (when sport and neighbourhood closures were in effect; March 2020–June 2020; January 2021–May 2021)?*” (During periods of lockdown in Ontario, various closures were in effect. For an extensive list COVID-19 related restrictions enforced since 2020 by time (e.g., month), please visit: https://www.jdsupra.com/legalnews/ontario-s-covid-19-response-a-history-1280608/ accessed on 29 April 2022). and “*In your opinion, how many minutes per day is your child engaging in physical activity currently (i.e., at this moment in time)?*”. Response options were consistent with the baseline survey, and a dichotomous variable was calculated for each time point (1 = engaging in 60 min of physical activity or more per day, 0 = engaging in less than 60 min of physical activity per day) [1].

### 2.6. Data Preparation and Analysis

All data preparation (e.g., cleaning) and analyses were completed in SPSS (version 28; [29]) and R studio using the lme4 [30] and lmerTest [31] packages. Incomplete survey responses (i.e., participants with more than 15% missing data from either survey) were removed [32]. Descriptive statistics, including means, standard deviations, and proportions, were used to report participant demographics. The proportion of children accumulating 60 min of physical activity at each time point was calculated from raw data. First, generalized linear mixed effects models with a binomial distribution and a logit link function were used to estimate changes in the proportion of children who accumulated 60 min of physical activity per day pre-lockdown, during lockdown, and post-lockdown. Time (i.e., pre-lockdown, during lockdown, post-lockdown) was entered as a factor to compare the proportion of children engaging in 60 min of physical activity per day at each time point. Second, to determine whether changes in physical activity during COVID-19 were moderated by sociodemographic characteristics, interaction effects were entered into the model between time and each of the measured sociodemographic variables (e.g., household income, level of education, dog ownership). Each of the demographic variables was dichotomized to aid interpretation of the moderation effect. Therefore, a three (time points) by two (dichotomized levels of the moderators) moderation analysis was conducted. All models were run with a random intercept to account for clustering of repeated measures within children. Given a small average cluster size (i.e., 1.68 children), clustering of children within families was not accounted for in the analyses. 

## 3. Results

A total of 243 parents (M_age_ = 38.8; SD = 5.7) caring for 408 children (M_age_ = 6.3 years; SD = 3.7) completed both online surveys. Most parents identified as female (94%), Caucasian (86%), had an average household income of CAD $100,000 or more (65%) and lived in a detached house (77%). Nearly half (41%) of participants reported their children’s biological sex as male, and 7% of children were reported to have a disability. See Table 1 for full participant demographics.

### 3.1. Change in Children’s Physical Activity during COVID-19

The proportions of children accumulating at least 60 min of physical activity per day, as per parent reports at each time point, are presented in Figure 2. Results from the generalized linear mixed effects model showed that children were significantly less likely to engage in 60 min of physical activity during lockdowns (20.53%) compared to pre-lockdowns (63.02%), *β* = −2.40, 95% CI = −2.83, −1.97, *p* < 0.001, Odds Ratio (OR) = 0.09, and during lockdowns compared to post-lockdowns (54.14%), *β* = −1.95, 95% CI = −2.36, −1.54, *p* < 0.001, OR = 0.14. Although the proportion of children engaging in 60 min of daily physical activity increased from during lockdowns to post-lockdown, children were still significantly less likely to engage in 60 min of physical activity per day post-lockdown compared to pre-lockdown (*β* = −0.46, 95% CI = −0.79, −0.12, *p* < 0.001, OR = 0.63).

### 3.2. Sociodemographic Moderators of Change in Children’s Physical Activity during COVID-19

Results from the moderation analysis are displayed in Table 2 and Figure 3. Results showed that the proportion of older children engaging in 60 min of physical activity per day decreased significantly more than younger children from pre-lockdowns to during lockdowns, *β* = −0.86, 95% CI = −1.66, −0.07, *p* = 0.033, OR = 0.42, as did the proportion among children from households with a family dog, *β* = −0.92, 95% CI = −1.72, −0.11, *p* = 0.025, OR = 0.40. Results also showed that the proportion of children from households with parents who are employed full-time to engage in 60 min of physical activity per day increased significantly more than children from households with parents who do not work full time from during lockdowns to post-lockdowns, *β* = 1.05, 95% CI = 0.27, 1.83, *p* = 0.008, OR = 2.86. Additionally, the proportion of children from non-double-parent households engaging in 60 min of physical activity per day decreased significantly more from pre-lockdown to post-lockdown than children from double-parent households, *β* = −1.33, 95% CI = −2.39, −0.27, *p* = 0.013, OR = 0.26, as did the proportion of females compared to males, *β* = −0.82, 95% CI = −1.57, −0.07, *p* = 0.032, OR = 0.44. Although not significant, results also demonstrated a U-shaped effect of household income on the proportion of children engaging in 60 min of physical activity per day, with a greater decrease from pre-lockdown to during lockdown in children from families with higher household income, then a rebound effect from during lockdowns to post-lockdown.

## 4. Discussion

This study examined Ontario children’s parent-reported physical activity levels over time throughout the COVID-19 pandemic (i.e., pre-lockdown, during lockdown, and post-lockdown). The proportion of children engaging in at least 60 min of physical activity per day, as reported by their parents, significantly declined during periods of lockdown in Ontario. However, children’s activity levels appeared to rebound after lockdowns were lifted, though they did not reach pre-pandemic levels [13,14]. Interestingly, older children showed a larger decline in physical activity levels during lockdown, as did children in households that had a family dog. Additionally, there was a significantly larger decrease in the proportion of children from non-double-parent households and girls engaging in 60 min of physical activity per day from pre-lockdown to post-lockdowns. Several findings are discussed below.

Given the many benefits of physical activity for children [33], exploring how participation has changed throughout the pandemic and during the implementation of specific public health measures is important. In addition, exploring whether levels of physical activity increased after lockdowns were lifted offers insightful considerations for the continuing and eventual post-pandemic recovery. Children accumulate physical activity in a variety of settings (e.g., school, sports, and community centers [34,35]), and many of these settings were closed or off limits during periods of the COVID-19 pandemic in Ontario. Additionally, as people were encouraged to remain at home, there were decreased opportunities for children to engage in incidental and spontaneous physical activities (e.g., on their way to school or while out with parents running errands). Therefore, it was not surprising that a significant decline in children’s physical activity was observed during periods when lockdown measures were in place. This decline in children’s physical activity during the COVID-19 pandemic is consistent with other studies [23,24,36]. For example, in a systematic review by Stockwell et al. [37] exploring changes in children’s and adolescents’ physical activity levels during the first year of COVID-19, all studies (*n* = 6) reported a decrease in children’s physical activity during the pandemic. Although the disruption of physical activity programming (e.g., after-school activities, sports) was necessary to reduce transmission of COVID-19, the noted decline in children’s physical activity may have been a result of children traditionally accumulating much of their physical activity in these settings [38], rather than at home. Prior to the start of the COVID-19 pandemic, Canadian data from early 2020 reported that 78% of 5–19 year old’s participated in some form of organized physical activity or sport [39]. In addition, 21% of this cohort also reported using active modes of transportation to travel to and from school [39]. With organized sports and schools closed for various periods during COVID-19 in Ontario, and therefore the need to commute to these settings negated, this could also have contributed to the identified decrease in children’s activity. 

Parents play an important role in the promotion of their children’s physical activity opportunities by role modeling, providing transportation, and financing activities [40,41]. Research has found that during the COVID-19 pandemic, many families have switched to more unstructured (e.g., playing outdoors) versus structured activity (e.g., organized sports [26,42]. As such, parents may play an even more crucial role in supporting their children’s movement behaviors, especially during stay-at-home periods, where activity stimulus or promotion is less likely to come from external sources (i.e., sports coaches, peers, and mentors). This may explain why results from the current study showed that children from non-double-parent households had a greater decrease in physical activity from pre-lockdown to post-lockdown than children from double-parent households, as single parents may juggle multiple household duties and may not have the time to support their children’s engagement in physical activity. Similar results have been reported in previous research, which showed that children in households with dual parents engaged in significantly more physical activity six months into the pandemic [24]. It is urgent that supports are put in place for single parents or for those who are reliant on structured movement opportunities to ensure that these children are getting sufficient activity when sports and organized activities are not accessible. Recent research shows that adults (including parents) have also reported that their activity levels have declined because of COVID-19 [37], suggesting that there may be a family effect of the pandemic on physical activity behaviors. Efforts to increase children’s physical activity should be targeted at the individual level by parents and ideally should be centered around how their children like to accumulate activity (whether at home, in unstructured settings such as outdoor environments, or in organized settings such as sports). In addition, the role of schools and municipalities in promoting children’s activity outside of the home should not be overlooked.

The results from the study suggested that elementary-school-aged children (6–12 years) may have had a larger decrease in physical activity than younger children (0–5 years). These findings are consistent with other Canadian research, which demonstrated younger children’s levels of physical activity decreased less during lockdowns [23]. This could be due to the cumulative effect of a lack of active transport to school and loss of organized activities that children approaching adolescence experienced during periods when schools and organized sports were closed. Moreover, there was a larger decrease in the proportion of females achieving 60 min of physical activity from pre-lockdown to post-lockdown when compared to males. This finding is consistent with pre-COVID-19 literature, which has persistently identified girls to be less active than boys [28], and is also consistent with research conducted during the pandemic [23]. This may have occurred because of increased time at home to engage in screen-viewing, less social pressure from parents/guardians, and no access to organized sport. However, these factors are not limited to COVID-19. Further investigations are needed to identify how the pandemic has affected the physical activity levels of children of different ages and biological sex, and it is crucial that children are given ample opportunities to get active during early childhood to increase the chance that these healthy habits track into later life [43].

Socio-economic status was also associated with changes in physical activity during the COVID-19 pandemic. The effect of socio-economic status was mainly observed from during lockdown to post-lockdown. There was a greater increase in the proportion of children from high-income families and families with parents who work full time from during lockdown to post-lockdown. Participation in organized sport and physical activities is greater among children from families with higher socio-economic status [44]. Therefore, these findings may be a consequence of a combined loss of sport and formalized physical activity opportunities during lockdown and a greater ability to reengage with formal sports and physical activities when they restarted following lockdown. 

Children from families with a dog also experienced greater declines in physical activity from pre-lockdown to during lockdowns. This is contradictory to previous research, which has largely shown a positive association between dog ownership and physical activity in youth [45], and research conducted in Canada during the COVID-19 pandemic, which found family dog ownership to be positively associated with healthy movement behaviors [23]. However, despite being associated with greater outdoor physical activity/play, this previous research also showed that dog ownership was associated with significantly less indoor play in Canadian youth [23]. We espouse that this decrease in physical activity may be more strongly attributed to the fact that lockdowns occurred during the winter months in Ontario (whereby seasonality was likely an issue [46]), and indoor play became more prominent, thus potentially diluting the impact of dog ownership on sustaining children’s activity in the present study. 

Considering the growing body of research exploring the impact of the COVID-19 pandemic on children’s movement behaviors, and findings that show alarmingly low percentages of children meeting guidelines during lockdowns [23,24], it is essential that more research be conducted to understand the supports needed for parents with young children. Specifically, support for parents is needed to ensure their children are meeting movement guidelines during periods when physical-activity-supportive environments are closed, such as in the case of more stay-at-home orders or following natural disasters. Although the results of the present study showed an upwards trend in the number of children achieving at least 60 min of physical activity post-lockdown, there remains work to be conducted to get children back to at least pre-pandemic levels of physical activity. Further, given that physical inactivity is associated with a wide array of chronic health conditions [8], increased supports for physically distanced activity are also needed. Finally, researchers in this area should aim to obtain objectively measured physical activity levels of this population, to allow for comparisons across studies. In addition, exploring the correlation between parents’ personal activity levels and their children’s levels is also warranted. 

### Strengths and Limitations

Strengths of the present study include the use of repeated measures to examine the impact of the COVID-19 pandemic on a provincial sample of parents and children and the use of generalized linear mixed effects models to present the non-linear trajectories of the proportion of children who accumulated at least 60 min of physical activity during various periods of the pandemic. Notwithstanding these strengths, there are some limitations to note. First, despite our efforts to retain as many participants as possible from baseline to follow-up, only 243 of the 819 participants at baseline completed the follow-up survey, limiting the size of our sample. Second, children’s physical activity was collected via parent report and did not capture intensity level (e.g., MVPA). This may have influenced our findings due to potential recall or social desirability biases [47] or led to participants reporting invalid data (e.g., lack of precision). Further, survey respondents were primarily female, Caucasian, with full-time employment, and had higher than average household incomes; this limits the generalizability of findings to other populations. Participants were also recruited by contacting sport organizations, which may also have led to parents of already physically active children participating. Additionally, it is important to note that public health measures may have differed across regions in Ontario, and this may have impacted parents’ reports of their children’s physical activity levels (depending on what was open at the time of survey completion in their respective city/town). Finally, this study did not capture parents’ nor children’s vaccination status and did not consider the impact this may have had on parents’ return to sport/play decision making.

## 5. Conclusions

This study explored changes in the proportion of children accumulating 60 min of daily physical activity during various periods of the COVID-19 pandemic in Ontario, Canada. It was found that children’s activity levels dropped during periods when the COVID-19 virus risk was high, and many public health protections were in place; however, findings should be interpreted with caution due to the nature of the study (e.g., self-reported physical activity levels) and high homogeneity of study participants (i.e., female, Caucasian, high household income). As society continues to navigate the changing landscape of the COVID-19 pandemic, it is imperative that we strive to understand whether the pandemic will have lasting impacts on physical activity levels for families with young children, and preparation is needed to support children’s movement behaviors through pandemic and non-pandemic times alike. 

## Figures and Tables

**Figure 1 children-10-00221-f001:**
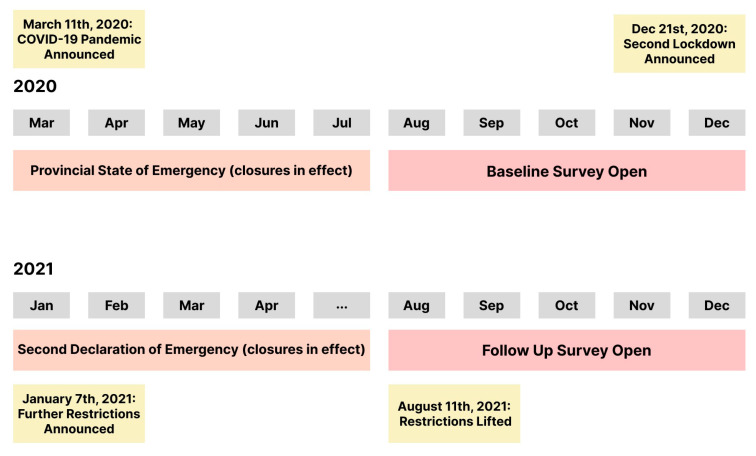
Timeline of the Return to Play Study Surveys and Provincial Mandates. *Note:* During provincial states of emergency (March–July 2020 and December 2020–August 2021), various closures and policies were in effect, including, but not limited to, stay-at-home periods, school closures, non-essential business closures (e.g., restaurants, gyms), and remote (work from home) work mandates for non-front line staff workers [15].

**Figure 2 children-10-00221-f002:**
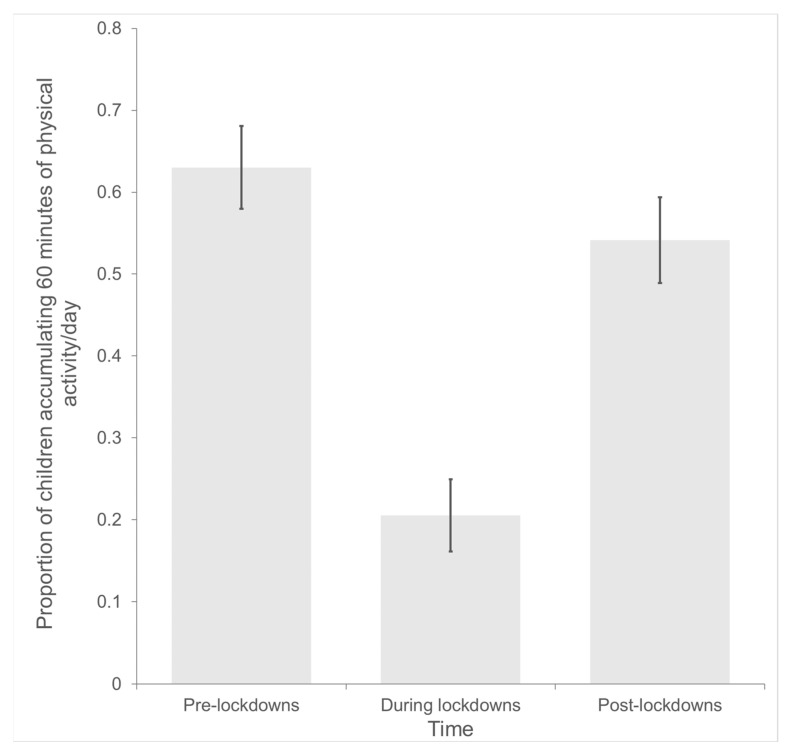
Proportion of children attaining 60 min of physical activity, as reported by parents, pre-lockdown, during lockdowns, and post-lockdown.

**Figure 3 children-10-00221-f003:**
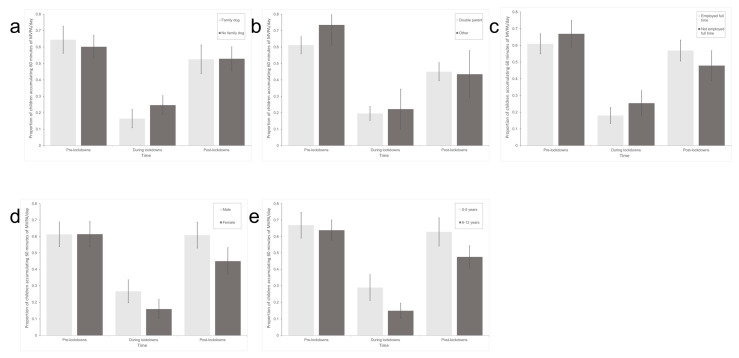
Difference in the proportion of children attaining 60 min of physical activity, as reported by parents, pre-lockdown, during lockdowns, and post-lockdown based on (**a**) owning a family dog, (**b**) family situation, (**c**) parent employ ent stats, (**d**) child sex, and (**e**) child age.

**Table 1 children-10-00221-t001:** Demographic Information of Parents (*n* = 243) and their Children (*n* = 408).

Parent Demographics	
**Gender**, *n* (%)	
Female	228 (93.83%)
Male	10 (4.12%)
**Age** (years), mean (***SD***)	38.76 (5.72)
**Ethnicity**, *n* (%)	
Caucasian	208 (85.60%)
South Asian	9 (3.70%)
East Asian	2 (0.82%)
Middle Eastern	1 (0.41%)
First Nations/Indigenous	7 (2.88%)
Latin American	3 (1.23%)
Other	6 (2.47%)
Prefer not to answer	3 (1.23%)
**Employment Status**, *n* (%)	
Full-Time	166 (68.31%)
Part-Time	30 (12.35%)
Unemployed	29 (11.93%)
Other	14 (5.76%)
**Family Situation**, *n* (%)	
Double-parent	203 (83.5%)
Single-parent	29 (11.9%)
Other	8 (2.9%)
**Housing Type**, *n* (%)	
Detached House	187 (76.95%)
Semi-Detached House	19 (7.82%)
Apartment/Condominium/Townhouse	31 (12.76%)
Other	2 (0.82%)
**Household Income**, *n* (%)	
Less than CAD $60,000	35 (14.6%)
CAD $60,000–99,999	48 (20.1%)
CAD $100,000+	156 (65.3%)
Prefer not to answer	15 (6.3%)
**Highest Level of Education**, *n* (%)	
High School	17 (7.00%)
College	52 (21.40%)
University	81 (33.33%)
Graduate School	89 (36.36%)
Prefer not to answer	4 (1.65%)
**Child Demographics**	
**Sex**, *n* (%)	
Male	167 (40.93%)
Female	161 (39.46%)
**Age** (years), mean (***SD***)	6.32 (3.66)
**Disability**, *n* (%)	
Yes	29 (7.11%)
No	368 (90.20%)

*Note:* Percentages do not add to 100% due to missing data.

**Table 2 children-10-00221-t002:** Influence of Sociodemographic Moderators on Changes in Parent-Reported Child Physical Activity Over Time During the COVID-19 Pandemic.

	Change from Pre-Lockdown to during Lockdown		Change from during Lockdown to Post-Lockdown		Change from Pre-Lockdown to Post-Lockdown	
Moderator	Moderation Effect	*p*-Value	Moderation Effect	*p*-Value	Moderation Effect	*p*-Value
Community type (0 = rural; 1 = (sub)urban)	0.86 (−0.08, 1.80)	0.073	−0.48 (−1.41, 0.45)	0.310	0.38 (−0.44, 1.17)	0.369
Housing type (0 = other; 1 = detached)	0.81 (−0.17, 1.79)	0.107	−0.68 (−1.37, 0.60)	0.171	0.12 (−0.71, 0.96)	0.776
Family dog (0 = no; 1 = yes)	−0.92 (−1.72, −0.11)	0.025°	0.65 (−0.15, 1.45)	0.112	−0.27 (−0.99, 0.46)	0.470
Household income (0 = <CAD $100,000; 1 = CAD $100,000+)	−0.65 (−1.40, 0.10)	0.093	0.74 (−0.02, 1.50)	0.055	0.10 (0.60, 0.79)	0.788
Family situation (0 = other; 1 = double parent)	0.43 (−0.70, 1.57)	0.455	0.90 (−0.21, 2.01)	0.113	1.33 (0.28, 2.38)	0.013
Parent employment status (0 = not employed full time; 1 = employed full time)	−0.21 (−0.98, 0.56)	0.588	1.05 (0.27, 1.83)	0.008°	0.84 (0.11, 1.57)	0.024
Parent education (0 = high school/college diploma, 1= university degree)	−0.20 (−0.97, 0.57)	0.629	0.46 (−0.35, 1.27)	0.262	0.26 (−0.48, 1.01)	0.485
Child sex (0 = female; 1= male)	0.74 (−0.08, 1.55)	0.077	0.09 (−0.74, 0.91)	0.838	0.82 (0.07, 1.58)	0.032
Child age (0 = 0–5 years; 1 = 6–12 years)	−0.86 (−1.66, −0.07)	0.033°	0.23 (−0.57, 1.03)	0.572	−0.63 (−1.37, 0.11)	0.094
Child disability (0 = no; 1 = yes)	−0.78 (−2.41, 0.84)	0.345	0.63 (−1.01, 2.27)	0.450	−0.15 (−1.47, 1.16)	0.820

***Note:*** ° *p* < 0.050.

## Data Availability

The data from the larger Return to Play is available on Western University’s Dataverse data repository. doi:10.5683/SP3/ZPWDR3.

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
