# Peer review of "Parent-Reported Changes in Ontario Children’s Physical Activity Levels during the COVID-19 Pandemic"

_children, 2023, doi:10.3390/children10020221_

Round 1

Reviewer 1 Report

- In the abstract, specify if survey 1 was pre-lockdown. Closures and pandemic restrictions varied between jurisdictions so it is important to establish contextual timelines.

-       Line 110-111, add statement related to collecting informed consent from participants

-       Line 122 onwards, suggest including a copy of the survey as a supplementary file to aid in transparency and reproducibility

-       Line 159, ‘missingness’ can likely be corrected to ‘missing’. Clarify what is meant by 15% missing. Is this 15% of total questions (including all question categories)?

-       Table 1. Clarify if parent sex or gender was collected.

-       Based on the data presented, it seems like some parents provided data for more than one child. Can you clarify how this was facilitated? Did each parent complete the survey more than once or were they asked if they were providing data representing more than one child (e.g., siblings who had the same activity level)? This is important to clarify in the procedures and analysis.

-       First sentence, second paragraph of discussion, suggest revising as it is quite long.

- The conclusions should be toned down to revised slightly to reflect the substantial limitations of this study (recall of physical activity, sample of participants with high socioeconomic status). 

Author Response

Thank you for taking the time to review our manuscript. We have addressed the comments and recommendations raised and feel that our paper is stronger because of the feedback received. Please see our point-by-point responses to each of your suggestions and comments below.

1) In the abstract, specify if survey 1 was pre-lockdown. Closures and pandemic restrictions varied between jurisdictions, so it is important to establish contextual timelines.

Response: The baseline survey (survey 1) was open from August – December 2020, and this can be seen in lines 27-29 in the abstract that state “Parents (n = 243; Mage = 38.8 years) of children aged 12 and under (n = 408; Mage = 6.7 years) living in Ontario, Canada completed two online surveys between August and December 2020 [survey 1] and August and December 2021 [survey 2].”. Because the COVID-19 pandemic was declared a few months prior, in March of 2020, survey 1 was launched during lockdown, and participants could complete the survey at any point during the abovementioned time (August – December 2020). In survey 1, participants were asked to report on their children’s physical activity levels retrospectively. Because lockdowns across Ontario differed during this time (August – December 2020) based on region, we have chosen not to include this information in the abstract as it requires more space to describe. However, we have presented this important contextual information that we believe you are looking for in our introduction section of our paper. Please see this on page 2, lines 91-94 that read, “Specifically, children’s physical activity levels prior to the declaration of a pandemic and lockdown (before March 2020; measured retrospectively), during lockdowns (March 2020- June 2020; January 2021 - May 2021), and post-lockdown (between August to December 2021) were captured*.In addition, we included a footnote (at the end of this statement, documented by the * symbol) to provide our readers with more information regarding the breakdown of COVID-19 restrictions in Ontario during the timelines of our study. Please see the footnote that reads “At post-lockdown, stay-at-home periods in Ontario ceased and various sport settings were permitted to re-open. For more information, a timely breakdown of COVID-19 restriction easing (e.g., settings that were open for operation) can be found here: https://www.jdsupra.com/legalnews/ontario-s-covid-19-response-a-history-1280608/”. We have also included a timeline of our study surveys that we believe will aid with understanding COVID-19 in Ontario during our study (Figure 1, page 3). We hope these additions address your concern.

2) Line 110-111, add statement related to collecting informed consent from participants

Response: Thank you for this suggestion. Please see our addition under the heading “Study Design and Procedures” on lines 111-114 that now reads “Ethical approval was provided by the Non-Medical Research Ethics Board at the University of Western Ontario (REB #116331), and informed consent was obtained from all participants.” In addition, we also provide this information under our header, “Recruitment and Participants” - please see lines 118-119 that read “Upon confirmation of study eligibility, participants were directed to the first online survey (the completion of which indicated their consent to participate).

3) Line 122 onwards, suggest including a copy of the survey as a supplementary file to aid in transparency and reproducibility

Response: Thank you for this suggestion. Please see the supplementary file that has been uploaded as a part of our revision for a copy of the survey. In addition, we have made note of this supplementary file under our Study Design and Procedures section (page 3, lines 111-112). In this supplementary file, we have included a copy of the components of the baseline survey used for this study – specifically the demographic questionnaire, as well as the items collecting physical activity related data. Items from the follow-up survey that were used in this study have also been included (e.g., physical activity levels). We did not include other items from the survey (i.e., those that were not used for the present study) in our supplementary file as the online surveys were very long, and other survey sections are not relevant to the present manuscript.

4) Line 159, ‘missingness’ can likely be corrected to ‘missing’. Clarify what is meant by 15% missing. Is this 15% of total questions (including all question categories)?

Response: 15% missing refers to the percentage of the survey that was not completed by the participant. In other words, if the participant did not complete at least 85% of the survey (in its entirety, regardless of survey section), their response was not included in data analysis. We have changed the language on page 5, line 182-183 from missingness to missing and added contextual information. Please see this on lines 182-183 that now read “Incomplete survey responses (i.e., participants with more than 15% missing data from either survey) were removed[32].”

5) Table 1. Clarify if parent sex or gender was collected.

Response: Parent gender was collected, and children’s biological sex (i.e., sex at birth) was collected. This information can be seen in the headers in Table 1. The demographic questions used to collect this information (i.e., parent gender and child biological sex) can also be seen in the supplementary file (online survey; baseline) that we have included alongside this study as per your recommendation.

6) Based on the data presented, it seems like some parents provided data for more than one child. Can you clarify how this was facilitated? Did each parent complete the survey more than once or were they asked if they were providing data representing more than one child (e.g., siblings who had the same activity level)? This is important to clarify in the procedures and analysis.

Response: That is correct. Parents were asked to provide data for all their children aged 12 and under in both online surveys. More specifically, In the baseline survey – demographics section, participants were asked how many children they had aged 12 or under. A dropdown list was used to collect responses [a function in Qualtrics online survey software], and participants could select any number (e.g., 2 if they had 2 children under 12 years of age at the time of survey completion). Based on the participants’ response to this question, the remainder of the survey was auto populated. For example, if a participant indicated that they had 2 children under 12 years of age, they were asked to complete the survey and provide responses for 2 children, from youngest to oldest (e.g., participants had to provide their children’s age and biological sex, physical activity, etc. for both children). We have provided more information on these details on page 4, under “Sociodemographic Questions” on lines 150 – 156. Please see our insertion that states “Participants were asked to report the number of children aged 12 and under they cared for at the time of survey completion (i.e., How many children aged 0-12 years do you currently provide care for?) via a dropdown list (i.e., participants could select any number). As such, the questions that followed were populated based on participants’ selected number of children. For example, if a participant selected that they have 2 children under 12 years of age, they had to fill out age and biological sex twice, once per each of their children.” We have also referred to this in the subsequent section “Children’s Physical Activity” to clarify that participants were required to report the physical activity levels of each of their children (based on the number of children they reported to have in their demographic section). Please see this on page 5, lines 161-166 that read “Specifically, the survey asked, “In your opinion, how many minutes per day did your child spend engaged in physical activity prior to the COVID-19 pandemic?” Response options were presented in multiple choice format, with four response options (i.e., less than 30 minutes per day, 30-59 minute per day, 60-149 minutes per day, or 150 minutes or more per day), and participants were required to input a value for each of their children.” With regards to statistical analysis, parents reported on average having 1.68 children (page 5; Line 199). Given the small average size of each cluster, it was not possible to control for the clustering of children within families in the analysis (see line 198). Therefore, all children were analyzed independently.

7) First sentence, second paragraph of discussion, suggest revising as it is quite long.

Response: Thank you for this suggestion. We have revised the sentence and believe this change is beneficial to the flow of our paragraph. Please see this change to the sentence in our discussion (lines 240-244) that now reads, “Given the many benefits of physical activity for children[33], exploring how participation has changed throughout the pandemic and during implementation of specific public health measures is important. In addition, exploring whether levels of physical activity increased after lockdowns were lifted offers insightful considerations for the continuing and eventual post-pandemic recovery.” We hope you that our re-structuring of this sentence makes it easier to read.

8) The conclusions should be toned down to revised slightly to reflect the substantial limitations of this study (recall of physical activity, sample of participants with high socioeconomic status). 

Response: We have added a statement to the conclusion to reflect the limitations of the study. Please see these changes made under Conclusion, lines 371-375 that read “It was found that children’s activity levels dropped during periods when the COVID-19 virus risk was high and many public health protections were in place; however, findings should be interpreted with caution due to the nature of the study (e.g., self-reported physical activity levels) and high homogeneity of study participants (i.e., female, Caucasian, high household income).” We have also removed the future directions from this section to tone down the conclusion statement. We hope that these changes satisfy your concern.

Reviewer 2 Report

With regard to manuscript: Parent-Reported Changes in Ontario Children’s Physical Activity Levels During the COVID-19 Pandemic, for consideration in Children. This is a very interesting manuscript addressing the physical activity participation during the COVID-19 pandemic. The paper will contribute to knowledge and is worthy of publication. I have comments and questions that I have detailed below.

·  The novelty of the study could be more highlighted (since already known the decrease in children’s physical activity during the pandemic). No similar attempt has been made by other countries around the world? The authors would explain why their findings aggregates to the existing knowledge.

·  Evaluations of Physical Activity were questionnaire-based. One caveat for the use of questionnaire is that they cannot capture all of the nuances of day-to-day life and do not reflect with precision real-life situations. I understand the difficult for collecting PA information in humans, however I strongly believe that these limitations (about use of questionnaire) should be assumed.

·  There is a necessity to explain the Moderation effect in table 2. how is it calculated and interpreted (this is crucial to the reader)? Please refine your manuscript.

·  The results should be more explored. As suggestion: To perform the absolute and percentage variation (∆) among groups. It would be interesting report the effect size (Cohen's d).

·  Replace the future investigations from the conclusions.

·  Describe the experimental conditions in more detail. About data collection, give more details on place of evaluations (on-line settings), time of day, instructions, time in each individual evaluation and others minor things.

·  It is unclear how proportion of children attaining 60 minutes of physical activity (fig1) were calculated. This should make it easy to the reader.

·  It is known that the world was surprised by the Coronavirus outbreak, forcing authorities to impose lockdown measures as school and workplace closings. These strategies against the COVID-19 pandemic (inducing children to stay at home) has dramatically changed not only their engagement in volitional activities (e.g. time spent in sports and planned exercise), but also the nonvolitional activities within the scope of spontaneous physical activity (SPA), which could be appreciated. Please see this review (https://pubmed.ncbi.nlm.nih.gov/21177942/) DOI: 10.1242/jeb.048397. I would like this point of view to be more in-depth in discussion.

·  Didactics would improve with the inclusion a figure in discussion (some scheme drawn by the authors) explaining their findings.

Author Response

Thank you for taking the time to review our manuscript. We feel that our paper is stronger because of your insightful contributions. Please see our point-by-point responses to each of your comments and suggestions below. 

1) This is a very interesting manuscript addressing the physical activity participation during the COVID-19 pandemic. The paper will contribute to knowledge and is worthy of publication. I have comments and questions that I have detailed below.

Response: Thank you for the positive note regarding our work, and for taking the time to review our manuscript.

2) The novelty of the study could be more highlighted (since already known the decrease in children’s physical activity during the pandemic). No similar attempt has been made by other countries around the world? The authors would explain why their findings aggregates to the existing knowledge.

Response: Thank you for bringing this to our attention. In our introduction, we highlight that previous research has identified the decrease in physical activity that occurred because of the pandemic (page 2, lines 64-74, and lines 77-81) to explicitly acknowledge that similar studies have been conducted on this topic. However, to express the novelty and contribution of our study to the literature, we specify that there is limited published research on how COVID-19 impacted the physical activity of children aged 0-5 (infants, toddlers, and preschoolers). We included children aged 0-5 in our study, as this has not been done before (to our knowledge). This can be found on page 2 of our paper, lines 87-90, that read “Given the lack of published evidence pertaining to the impact of the pandemic on young children’s (0-5 years) movement behaviours, the purpose of this study was to explore changes in Ontario children’s (0-12 years) physical activity levels during the first 1.5 years of the COVID-19 pandemic.” In addition, although other studies have demonstrated a decrease in physical activity, limited research exists on the factors that explain this decrease (a gap that our study fills). Specifically, our study explored the family sociodemographic and environmental factors that may help to understand the decrease in physical activity levels. We have explained in this in our discussion (described as the secondary objective of this study – as seen in lines 96-99). We hope this identifies the unique aspect of our research question, our efforts to explain what our study adds to the existing knowledge (as seen in our discussion section) and our contribution to the literature.

3) Evaluations of Physical Activity were questionnaire-based. One caveat for the use of questionnaire is that they cannot capture all of the nuances of day-to-day life and do not reflect with precision real-life situations. I understand the difficulty for collecting PA information in humans, however I strongly believe that these limitations (about use of questionnaire) should be assumed.

Response: We agree that there are limitations to assessing physical activity levels via a questionnaire. We have acknowledged this in our Strengths and Limitations section of our paper (lines 348-367) and have inserted contextual information about this limitation. Please see these changes on lines 355-358, “Second, children’s physical activity was collected via parent-report, and did not capture intensity level (e.g., MVPA). This may have influenced our findings due to potential recall or social desirability biases[47], or led to participants reporting invalid data (e.g., lack of precision).” We hope these additions show our effort to be as transparent as possible with regard to our study limitations.

4) There is a necessity to explain the Moderation effect in table 2. How is it calculated and interpreted (this is crucial to the reader)? Please refine your manuscript.

Response: We have clarified that interaction terms were entered between time (i.e., pre-lockdown, during-lockdown, post-lockdown) and each of the socio-demographic variables. Each demographic variable was dichotomized to aid interpretation. We have added this information to the manuscript to our data preparation and analysis section (page 5; lines 195-198).

5) The results should be more explored. As suggestion: To perform the absolute and percentage variation (∆) among groups. It would be interesting report the effect size (Cohen's d).

Response: We have reported on the absolute percentage and provided plots of the proportion of children participating in 60 minutes of physical activity per day at each time point. We have now also provided plots of the proportion of children accumulating 60 minutes of physical activity per day at each time point for subgroups that significantly moderated changes in physical activity over time. Lastly, we have reported odds ratios as an effect size. Please see these additions on lines 230-248. Please also see the addition of Figure 3 that visually displays the difference in the proportion of children attaining 60 minutes of physical activity, as reported by parents, pre-lockdown, during lockdowns, and post-lockdown based on owning a family dog, family situation, parent employment stats, child sex, and child age.

6) Replace the future investigations from the conclusions.

Response: We have removed the future directions from the conclusions and moved it to the end of our discussion section. Please see this on lines 340-346 that read “Further, given that physical inactivity is associated with a wide array of chronic health conditions[8], increased supports for physically distanced activity are also needed. Finally, researchers in this area should aim to obtain objectively measured physical activity levels of this population, to allow for comparisons across studies. In addition, exploring the correlation between parents’ personal activity levels and their children’s levels is also warranted.

7) Describe the experimental conditions in more detail. About data collection, give more details on place of evaluations (on-line settings), time of day, instructions, time in each individual evaluation and others minor things.

Response: Both of our online surveys were open for 5-month periods (i.e., August – December 2020 and August – December 2021). As such, participants were able to complete the surveys at any convenient time for them during this period. There were no rules regarding time of day that the survey had to be completed, or how much time participants were able to spend on the survey. In our letter of information, we outlined that the surveys would take approximately 30 minutes to complete. We have inserted this contextual information of the experimental conditions in our methodology section. Please see this on page 4, lines 134 – 139 that now state “Two online surveys were created by the research team, and the Tolerance for Risk in Play Scale (TRiPS) validated tool was incorporated into the baseline survey[27]. The surveys were administered in English and delivered via Qualtrics and took approximately 30 minutes to complete. Survey items were informed by the COVID-19 situation in Ontario, Canada at the time of survey creation (i.e., survey 1, launched August 2020, n = 162 items; and survey 2, launched August 2021, n = 58 items). Survey items were by embedded into sections based on type of question asked (e.g., demographics, physical activity levels), and unique instructions were provided based on each section.” Thank you for this suggestion as we feel this information is important to provide to our readers.

8) It is unclear how proportion of children attaining 60 minutes of physical activity(fig1) were calculated. This should make it easy to the reader.

Response: We have clarified that the proportion of children attaining 60 minutes of physical activity were calculated from raw data. The 95% confidence intervals were generated using the equation p + 1.96 * sqrt([p(1-p)]/n) where p is the proportion and n is the sample size. We have made this information clear in the Data Preparation and Analysis section on page 5, lines 185-186.

9) It is known that the world was surprised by the Coronavirus outbreak, forcing authorities to impose lockdown measures as school and workplace closings. These strategies against the COVID-19 pandemic (inducing children to stay at home) has dramatically changed not only their engagement in volitional activities (e.g. time spent in sports and planned exercise), but also the nonvolitional activities within the scope of spontaneous physical activity (SPA), which could be appreciated. Please see this review (https://pubmed.ncbi.nlm.nih.gov/21177942/) DOI: 10.1242/jeb.048397. I would like this point of view to be more in-depth in discussion.

Response: We thank the author for their recommendation and have now alluded to the decreased opportunity to engage in incidental and spontaneous physical activities in the discussion section of our paper. This added text reads: “Additionally, as people were encouraged to remain at home there were decreased opportunities for children to engage in incidental and spontaneous physical actives (e.g., on their way to school or while out with parents running errands).” Please find this on lines 246-249.

10) Didactics would improve with the inclusion of a figure in discussion (some scheme drawn by the authors) explaining their findings.

Response: We thank the reviewer for the suggestion, although we disagree with their suggestion of including a figure in the discussion. We have now included additional figures in the results section of the manuscript to further aid with the interpretation of the study finding.

Reviewer 3 Report

1. Although the authors defined the data collection phases as pre-, during, and post-pandemic, the COVID-19 policies were not mentioned clearly in the paper so the readers cannot know what changes happened among families in the study area. Please provide more information about this.

2. The authors described that they used Generalized linear mixed effects models to analyze the repeated designs of the study. However, I'm not sure whether the moderation mentioned is appropriate or clear enough in this method since the outcomes were changes from pre- to during (and others) as two-time points. Please provide more illustrations about what are the steps and analytical plans you conducted. 
3. Past research has suggested that self-reported physical activity time is not reliable. Please justify the use of the measures.

Author Response

Thank you for taking the time to review our manuscript. We have addressed all of your comments, and feel that our paper is stronger because of your contributions. Please see our point-by-point response to each of your comments below.

1) Although the authors defined the data collection phases as pre-, during, and post-pandemic, the COVID-19 policies were not mentioned clearly in the paper so the readers cannot know what changes happened among families in the study area. Please provide more information about this.

Response: Thank you for bringing this to our attention, and we apologize that this was not clear. We defined the data collection phases as pre- during and post-lockdown because we felt this was the most straightforward definition of our time points. For example, pre-lockdown referred to time before the onset of COVID-19 in Ontario (specifically, before March 2020 when COVID was declared a global pandemic). We believe that this information is presented very clearly in our methods section, under the heading “Children’s Physical Activity”. Please see 160-66 that read “In Survey 1, parents reported their children’s daily physical activity levels pre-lockdown. Specifically, the survey asked, “In your opinion, how many minutes per day did your child spend engaged in physical activity prior to the COVID-19 pandemic?””. For during and post-lockdown, this information can be found on lines 169-179 that read “At follow-up (i.e., Survey 2), parents responded to two questions concerning each of their children’s physical activity during lockdowns and post-lockdown. The questions asked, “In your opinion, how many minutes per day did your child spend engaged in physical activity during Ontario’s strictest COVID-19 related lockdowns (when sport and neighbourhood closures were in effect; March 2020- June 2020; January 2021 - May 2021)?” and “In your opinion, how many minutes per day is your child engaging in physical activity currently (i.e., at this moment in time)?”.  We would also like to point out that in our introduction we provide more contextual information regarding COVID-19 and public health measures in Ontario during the timeframe of our study and have included a figure (figure 1) to aid in our readers’ understanding of COVID-19 timelines and policies in Ontario during our study period. We have also inserted a footnote in our introduction with a supporting reference that directs our readers to a resource that explains phases of re-opening that were put in place during the first year of COVID-19 (2020-2021; timing of this study). The resource also provides a specific breakdown of what types of settings were open during each phase, and how this varied by city in Ontario. Please see this on page 2, lines 91-94, where we define our purpose statement of this study and describe the data collection phases – “Specifically, children’s physical activity levels prior to the declaration of a pandemic and lockdown (before March 2020; measured retrospectively), during lockdowns (March 2020- June 2020; January 2021 - May 2021), and post-lockdown (between August to December 2021) were captured*.The referenced footnote (specified by the * symbol) reads: At post-lockdown, stay-at-home periods in Ontario ceased and various sport settings were permitted to re-open. For more information, a timely breakdown of COVID-19 restriction easing (e.g., settings that were open for operation) can be found here: https://www.jdsupra.com/legalnews/ontario-s-covid-19-response-a-history-1280608/

2) The authors described that they used Generalized linear mixed effects models to analyze the repeated designs of the study. However, I'm not sure whether the moderation mentioned is appropriate or clear enough in this method since the outcomes were changes from pre- to during (and others) as two-time points. Please provide more illustrations about what are the steps and analytical plans you conducted. 

Response: We conducted two analyses. First, we used general linear mixed models to examine how the proportion of children accumulating 60 minutes of MVPA each day differed at each time point (i.e., main effect). Second, we conducted moderation analysis to determine whether socio-demographic factors moderated changes in physical activity over time (i.e., interaction effects). We have added additional information to the Data Preparation and Analysis section of the manuscript to clarify this point (page 5).

3) Past research has suggested that self-reported physical activity time is not reliable. Please justify the use of the measures.

Response: Due to the nature of the pandemic in Ontario (e.g., physical distancing requirements), and our eagerness to launch this study in a timely manner (e.g., during initial stages of the pandemic), we opted for parent self-report physical activity measures that could be collected via our online survey. Further, it would not have been feasible for us to collect physical activity data via objective measures such as via accelerometers in the same timeline as presented in the current paper, due to the wide geographical spread of our participants (e.g., across Ontario). This is because of our inability to travel to participants to measure their activity objectively during initial phases of COVID-19 in Ontario, not only because of the time commitment, but also because of the public health measures that were in place at the time of data collection. We acknowledge that using self-report to measure the physical activity levels of our participants is a limitation of our study, and this has been noted in our limitation section. Further, we have also noted a common limitation of self-report measures, and that is social desirability bias. Please see this statement on lines 355-359 under strengths and limitations that read “Second, children’s physical activity was collected via parent-report, and did not capture intensity level (e.g., MVPA). This may have influenced findings due to potential recall or social desirability biases[47].” We hope this justifies our use of the measures and identifies how we have provided this information in our manuscript to be as transparent as possible about the limitations.

Round 2

Reviewer 1 Report

Thank you for addressing all my concerns. 

Reviewer 3 Report

The authors have addressed all my questions.